# Effects of Sodium Selenite on Accumulations of Selenium and GABA, Phenolic Profiles, and Antioxidant Activity of Foxtail Millet During Germination

**DOI:** 10.3390/foods13233916

**Published:** 2024-12-04

**Authors:** Shuaiduo Sun, Jingjing Zhang, Yongji Li, Yunfeng Xu, Runqiang Yang, Lei Luo, Jinle Xiang

**Affiliations:** 1Faculty of Food & Bioengineering, Henan University of Science & Technology, Luoyang 471023, China; 15037912413@163.com (S.S.); z2823662022@163.com (J.Z.); 15539578706@163.com (Y.L.); xuyunfeng1573@126.com (Y.X.); 13623896431@139.com (L.L.); 2Henan International Joint Laboratory of Food Green Processing and Safety Control, Henan University of Science & Technology, Luoyang 471023, China; 3College of Food Science & Technology, Nanjing Agricultural University, Nanjing 210000, China; yangrq@njau.edu.cn

**Keywords:** selenium, foxtail millet sprouts, polyphenols, GABA, antioxidant activity

## Abstract

This study investigated the influence of soaking and spraying with a sodium selenite (Na_2_SeO_3_) solution on selenium accumulation, γ-aminobutyric acid (GABA) content, phenolic compositions, and the antioxidant activity of foxtail millet sprouts. The screening results showed that foxtail millet seeds soaked with 60 mg/L of Na_2_SeO_3_ solution and sprayed with 2 mg/L of Na_2_SeO_3_ solution were the appropriate concentrations for the germination process. Compared with the spraying method, a presoaking treatment presented far higher selenium content and significantly higher (*p* < 0.05) selenium enrichment rates in foxtail millet sprouts. The content of free and bound phenolics, as well as GABA, were significantly (*p* < 0.05) increased in foxtail millet sprouts through both soaking and spraying treatments. Correspondingly, most of the individual phenolic compounds were significantly (*p* < 0.05) increased, especially after germination for 3 days. *Trans*-ferulic acid and *trans*-*p*-coumaric acid were the predominate bound phenolic acids, feruloylquinic acid and 4-*p*-coumaroylquinic acid were the major free-form phenolic compounds, and *N*-feruloyl serotonin and *N*-(*p*-coumaroyl) serotonin were the new arising phenolic derivatives caused by germination. Both the soaking and spraying treatments induced the enrichment of these individual phenolic compositions, thus increasing the total phenolic content and in vitro antioxidant activity of foxtail millet sprouts. It was indicated that selenium-enriched germination treatment should be an effective method to produce functional selenium-enriched foxtail millet sprouts with more abundant GABA and polyphenols, thus enhancing the health benefits and added value of foxtail millet.

## 1. Introduction

Selenium (Se) is recognized as a vital trace element crucial for human health [1]. It is estimated that approximately one billion individuals globally experience inadequate selenium intake [2,3]. Se deficiency often leads to declines in cognitive ability, oxidative stress, decreased adaptive immunity, neurological disorders, and Keshan disease [4]. Edible plants are believed to be the main dietary source of Se for human, since many plants can metabolize and accumulate organic Se in their edible parts to be consumed directly, such as the leaves, flowers, fruits, seeds, and sprouts [5,6]. However, the Se content of food resources, especially organic Se, is generally very low due to the imbalance of Se in the soil of different regions [7]. Increasing the concentration of absorbable Se in edible plants or their products by Se-biofortification is a viable pathway to supplement Se for humans [3].

Se-biofortification in grain seeds through germination is believed to be a simple, rapid, and efficient method for the development of functional Se-enriched foods [8]. The germination process has also been widely applied as a low-cost and efficient method to enhance the nutritional value of grain seeds [9]. This method of Se-enriched sprouting has been proven to greatly improve the Se concentration in germinated seeds, such as mung bean sprouts [10], alfalfa sprouts [11], pea sprouts [12], and corn sprouts [13]. Additionally, Se-biofortification could also lead to accumulations of amounts of bioactive secondary metabolites during germination [5]. It has been reported that maize seeds presoaked with 50 mg/L of Na_2_SeO_3_ and germinated for 2 days displayed a 181% increase in bound ferulic acid when compared to germinated maize without Na_2_SeO_3_ treatment [14]. Se-biofortification increased the phenolic acid contents of rice sprouts, especially their soluble conjugated forms [15]. Anthocyanins were effectively enhanced by spraying radish shoots with 10 µM of Na_2_SeO_3_ solution during germination [16]. Compared with untreated chickpea sprouts, the total isoflavonoid content was increased by 83% through presoaking with 23.5 mg/L of Na_2_SeO_3_ solution [17]. The effects of Se-biofortification on other phytochemicals and nutrients have also been reported, such as chlorophyll and carotenoids [18,19], GABA [20], and vitamin C [19,20]. The accumulation of these bioactive compounds and the simultaneous enrichment of organic Se are conducive to the enhancement of health benefits for germinated grain seeds, which have the potential to be used for the development of functional foods [21].

Foxtail millet is one of the most important minor cereal crops, and it has been considered a suitable candidate for the development of functional foods [22]. Traditional and modern processing technologies, together with their resulting nutritional and functional changes to foxtail millets, have been previously reviewed [23]. In our previous study, the phenolic and GABA content and antioxidant capacity of foxtail millet were significantly increased by the germination process [24,25]. However, little research related to the Se-biofortification of cereal grain sprouts during germination has been reported, and the effects of Se-biofortification on the secondary metabolites of germinated foxtail millets remain unclear.

The main purpose of this study was to (1) screen the appropriate Na_2_SeO_3_ treatment method and concentration for the Se-biofortification of foxtail millet sprouts; and (2) investigate the effects of Se-biofortification on Se content, GABA concentration, phenolic profiles, and antioxidant activity during the Se-enriched germination of foxtail millet. This research could contribute to the development of Se-enriched functional foods and improve the utilization value of foxtail millet sprout as a functional ingredient.

## 2. Materials and Methods

### 2.1. Reagents and Standards

Sodium selenite (Na_2_SeO_3_) was purchased from Tianjin Chemical Reagent Third Factory Co., Ltd. (Tianjin, China). DAN (2,3-Diaminonaphthalene) was purchased from Shanghai Macklin Biochemical Technology Co., Ltd. (Shanghai, China). Cyclohexane was acquired from Tianjin Yongda Chemical Reagent Co., Ltd. (Tianjin, China). Folin–Ciocâlteu reagent, 6-hydroxy-2,5,7,8-tetramethylchromane-2-carboxylic acid (Trolox), 2,2′-diphenyl-1-picrylhydrazyl (DPPH), 2,2-azinobis-(3-ethylbenzothiazoline-6-sulfonate (ABTS), and TPTZ (2,4,6-tris(2-pyridyl)-s-triazine) were acquired from Sigma-Aldrich Chemical Co. (St. Louis, MO, USA). All qualitative and quantitative standard products, including γ-aminobutyric acid, L-phenylalanine, tryptophan, p-hydroxybenzaldehyde, p-hydroxybenzoic acid, protocatechuic aldehyde, protocatechuic acid, vanillic acid, syringic acid, p-coumaric acid, ferulic acid, kaempferol, caffeic acid, and rutin, were purchased from Shanghai Yuanye Biotechnology Co., Ltd., (Shanghai, China). Methanol, formic acid, and acetonitrile (chromatographic grade) were purchased from Thermo Fisher Scientific Reagent Co., Ltd., (Waltham, MA, USA). Other analytical grade chemicals and reagents, such as sodium hydroxide (NaOH), sodium carbonate (Na_2_CO_3_), sodium nitrite (NaNO_2_), aluminum chloride (AlCl_3_), ferric chloride (FeCl_3_), ethyl acetate, n-hexane, and methanol, were purchased from Tianjin Deen Chemical Reagent Co., Ltd., (Tianjing, China).

### 2.2. Foxtail Millet and Germination Procedure

Foxtail millet samples were provided by Jinsu Agricultural Technology Co., Ltd. (Luoyang, China). To screen the Na_2_SeO_3_ presoaking solution concentration, the millet seeds were selected after cleaning and washing with pure water and soaked in 0, 20, 40, 60, 80, and 100 mg/L of Na_2_SeO_3_ solution at 25 °C for 12 h. The presoaked seeds were evenly placed into petri dishes and then germinated in an incubator at 25 °C for 3 days, and sprayed with pure water every 12 h to keep moisture. To screen the Na_2_SeO_3_ spraying solution concentration, the cleaned millet seeds were soaked in distilled water at 25 °C for 12 h, and then germinated at 25 °C in the dark. Different concentrations of Na_2_SeO_3_ solution (0, 2, 4, 6, 8, and 10 mg/L) were sprayed every 12 h to maintain suitable germination moisture. For investigating the effects of Na_2_SeO_3_ treatments on foxtail millet sprouts, three groups of germination experiments were carried out. In the control group, foxtail millet seeds were germinated with pure water; in the Se soaking group, the seeds were germinated by soaking in Na_2_SeO_3_ solution; and in the Se spraying group, the seeds were sprayed with Na_2_SeO_3_ solution during the germination process. The germination process was carried out simultaneously for 4 days. Samples were taken every day and recorded as G1, G2, G3, and G4, respectively. The ungerminated raw seed samples were recorded as G0. The morphology of foxtail millet sprouts during germination under different treatments is displayed in Appendix A.

The collected sprouts were washed with pure water to wipe off residual Na_2_SeO_3_, and then freeze-dried. Subsequently, the samples were crushed, sieved, and stored at −20 °C before the following analyses.

### 2.3. Determination of Se Content

The total Se content was determined using fluorescence spectrophotometry (Cary Eclipse; Agilent Instruments, Santa Clara, CA, USA) [26]. Briefly, foxtail millet sprout powder (0.5 g) was thermally digested overnight with 10 mL of a mixed solution of HNO_3_ and HClO_4_ (*v*/*v*, 9:1), and then 5 mL of HCl (6 M) was added to the digested solution. The thermally digested sample was reacted with 2,3-diaminonaphthalene, and the fluorescence intensity was measured after cyclohexane extraction. A selenium standard solution was used to draw the calibration curve, and the total Se content was calculated in milligrams per kilogram (mg/kg).

The extraction of inorganic Se was conducted according to the method described by C. Cheng et al. [27], with slight modifications. In short, the millet sprout powder (0.2 g) was mixed with 20 mL of ultrapure water, and then ultrasonic treatment was conducted for 15 min. The supernatant was obtained through centrifugation at 1000 rpm for 20 min to obtain the inorganic Se extract. The inorganic Se content was determined by the same procedure for the measurement of the total Se.

The organic selenium content and enrichment rate of selenium were calculated using Equations (1) and (2), respectively.
O_se_ = T_se_ − I_se_(1)
R_se_ = O_se_/T _se_ × 100%(2)
where T_se_ represents the total selenium content of foxtail millet sprouts, O_se_ represents the organic selenium content, I_se_ represents the inorganic selenium content, and R_se_ represents the selenium enrichment rate.

### 2.4. Determination of Phenolic Content

The free and bound phenolic compounds of foxtail millet sprouts were extracted, and the total phenolic content (TPC) and the total flavonoid content (TFC) were determined by our previously reported methods [28]. The freeze-dried samples were defatted twice with hexane (1:10 *w*/*v*, 15 min) at room temperature, and then mixed with 80% methanol and extracted in a water bath constant temperature oscillator (SHA-C, Jintan district Chenghui instrument factory, Changzhou, China) for 1 h. The supernatant was collected through centrifugation. This extraction process was repeated twice to obtain the free phenolic extract. After this, the residue from the free polyphenolic extraction was air-dried, and then hydrolyzed with 2 mol/L of NaOH solution for 2 h under nitrogen. The pH of the resulting slurry was adjusted to a pH of 1.5–2.0 with 6 mol/L of HCl. The hydrolysate was obtained through centrifugation and then extracted three times with ethyl acetate to obtain the bound phenolic extract. All of the liquid free or bound fractions were combined separately and evaporated to dryness by using a rotary evaporator and then re-dissolved in a 50% methanol solution for further analysis.

The TPC of the free or bound extract was determined on a 96-well microplate (PT06-96S, BIOLAND Biotechnology Co., Ltd. Hangzhou, China) by using the Folin–Ciocalteu assay. In brief, 20 μL of the appropriately diluted phenolic extract was combined with 40 μL of the Folin–Ciocalteu reagent, and then 160 μL of sodium carbonate solution (Na_2_CO_3_, 75 g/L, *w*/*v*) was added. The mixture was reacted for 2 h in the dark at 25 °C. A standard curve was prepared using ferulic acid, and the TPC was expressed as the milligrams of ferulic acid per 100 g of the dry weight (mg FAE/100 g DW).

The TFC was determined through the AlCl_3_ colorimetric method. Briefly, the phenolic extract was mixed with 75 μL of NaNO_2_ (5 g/100 mL), 150 μL of AlCl_3_ (10 g/100 mL) was added after 5 min, and 0.5 mL of NaOH (4 g/100 mL) and pure water were then added to obtain a final volume of 2.5 mL. After reaction for 15 min, the absorbance was read at 510 nm. A standard curve was prepared using rutin, and the TFC was expressed as milligrams of rutin per 100 g of the dry weight (mg RE/100 g DW).

### 2.5. Determination of GABA Content

The extraction process for GABA was similar to that of the free phenolics. In brief, the defatted sample was mixed with 80% methanol solution to obtain the GABA extract. GABA quantitative analysis was performed on a UPLC (Waters UPLC H-Class) coupled to a QqQ-MS (Waters Xevo TQ-S/micro, Waters Corp., Milford, MA, USA), according to the previously described method of Li et al. [24]. The chromatography was performed on an Accucore C_18_ column (100 mm × 3 mm, Thermo Fisher Scientific, Waltham, MA, USA), with a 10 μL injection and 0.2 mL/min flow rate. The column temperature was set at 30 °C. The liquid elution procedure was as follows: 0–3 min, 95–5% B; 3–4 min, 10–90% B; 4–6 min, 10–90% B; and 6–8 min, 95–5% B, where A and B were the water containing 0.1% formic acid and acetonitrile solution, respectively. The mass spectrometry employed the Multiple Reaction Monitoring (MRM) mode under a positive ion (ESI^+^) condition, with qualitative and quantitative ion pairs being 104/69.1 (*m*/*z*) and 104/87.1 (*m*/*z*), respectively. The other mass spectrometry parameters were as follows: a capillary voltage of 0.35 KV, a cone voltage of 30 V, an ion source temperature of 150 °C, and a solvent gas temperature of 500 °C. The flow rate of the solvent gas (N_2_) was 800 L/h, and the flow rate of the cone gas (He) was 20 L/h. A quantitative calculation of the peak area was performed using external calibration curves to obtain the GABA content in the samples.

### 2.6. Identification and Quantification of Individual Polyphenols

A UPLC (Waters UPLC H-Class) coupled to a QqQ-MS (Waters Xevo TQ- S/micro) was applied for the qualitative and quantitative analysis of phenolic compounds. The equipped chromatographic column was the Accucore C_18_ column (100 mm × 3 mm, Thermo Fisher Scientific, Waltham, MA, USA). The liquid elution procedure and mass spectrometry parameters were set to be the same as the previously described method by Xiang et al. [29]. The liquid phase conditions were as follows: the binary mobile phases A and B were 0.1% formic acid water and 0.1% formic acid methanol, respectively. The column temperature was 30 °C, the injection volume was 3 μL, and the flow rate was set to 0.4 mL/min. A 25 min linear gradient was programmed as follows: 0–3.81 min, 9–14% B; 3.81–4.85 min, 14–15% B; 4.85–5.89 min, 15% B; 5.89–8.32 min, 15–17% B; 8.32–9.71 min, 17–19% B; 9.71–10.40 min, 19% B; 10.40–12.48 min, 19–26% B; 12.48–13.17 min, 26–28% B; 13.17–14.21 min, 28–35% B; 14.21–15.95 min, 35–40% B; 15.95–16.64 min, 40–48% B; 16.64–18.37 min 48–53% B; 18.37–22.53 min, 53–70% B; 22.53–22.88 min, 70–9% B; and 22.88–25.00 min, 9% B. The mass spectrum conditions were as follows: MS/MS information was collected in the negative ion mode with a scan range of 50–1000, a collision energy of 20 V, a cone voltage of 20 V, a capillary voltage of 1 kV, cone gas (He) with a flow rate of 50 L/h, desolvation gas (N_2_) with a flow rate of 1000 L/h, and a temperature of 500 °C.

For phenolic compounds with the available standard, a calculation was performed using an external calibration curve. The phenolic derivatives without available standards were quantified as the equivalent concentration of the closest analog. The results were expressed in milligrams per kilogram of the dry weight (mg/kg DW) of foxtail millet sprouts.

### 2.7. Determination of In Vitro Antioxidant Activity

The DPPH radical scavenging capacities were determined by the method of Zheng et al. [30]. In brief, 10 μL of the polyphenol extract was added to a 96-well microplate, along with 190 μL of DPPH radical solution. After incubating at room temperature for 30 min, the absorbance at 515 nm was measured.

The ABTS^+^ radical scavenging capacities were determined by the method of Xiang et al. [31]. In short, the polyphenol extract of 10 μL was combined with 190 μL of ABTS^+^ working liquid on a 96-well microporous plate. After incubation at 25 °C for 30 min, the absorbance at 750 nm was measured.

The ferric-reducing antioxidant potential (FRAP) was determined according to Yuan et al. [32]. In short, 10 μL of the polyphenol extract was mixed with 300 μL of the iron-TPTZ reagent. The absorbance was measured at 600 nm after the reaction at room temperature for 2 h.

The DPPH, ABTS^+^ radical scavenging activities, and FRAP of the polyphenol extract from foxtail millet sprouts were all expressed in micromoles of Trolox equivalents per gram (μmol TE/g). The DPPH, ABTS^+^ radical scavenging activities, and FRAP of the foxtail millet sprouts were the sum of the corresponding DPPH, ABTS, and FRAP values of the free and bound phenolics, respectively.

### 2.8. Statistical Analysis

All experiments were carried out in triplicate, and the results were expressed as the mean ± standard deviation (SD). Statistical analyses were performed by SPSS 27 (v. 27, IBM Corp., Armonk, NY, USA). Significant differences were analyzed by using Tukey’s test at a *p*< 0.05 level.

## 3. Results and Discussion

### 3.1. Screening of Presoaking Na_2_SeO_3_ Concentration for Se-Enriched Foxtail Millet Sprouts

#### 3.1.1. Effect of Na_2_SeO_3_ Presoaking Concentration on Se Content

The total Se, organic Se, inorganic Se content, and Se-enrichment rate of foxtail millet sprouts at different Na_2_SeO_3_ soaking concentrations are shown in Figure 1. The Na_2_SeO_3_ soaking treatment resulted in a significant (*p* < 0.05) increase in the total Se and organic Se content of foxtail millet sprouts, which increased with the increase in Na_2_SeO_3_ solution concentration. The total Se content of the Se-enriched foxtail millet sprouts ranged from 7.83 to 36.46 mg/kg of DW, the organic Se content ranged from 6.27 to 26.54 mg/kg, and the Se-enrichment rate varied from 72.79% to 80.43%. Organic Se was the primary form in the foxtail millet sprouts with Na_2_SeO_3_ presoaking treatment, indicating that millet sprouts can effectively transform and accumulate organic Se. Similar results were observed in mung beans germinated through Na_2_SeO_3_ soaking treatment, where the total and organic Se content of mung bean sprouts gradually increased with the increasing Na_2_SeO_3_ soaking concentration [33]. However, the Se-enrichment rate decreased gradually with the increase in Na_2_SeO_3_ soaking solution concentration in this study. This may be attributed to the decreased ability of selenite assimilation and transformation into organic Se when the Na_2_SeO_3_ concentration reached a certain threshold level. Similar results have been reported for Se-enriched soybean sprouts [34].

#### 3.1.2. Effects of Na_2_SeO_3_ Presoaking Concentration on Phenolic and GABA Contents

Foxtail millet is a rich source of phenolic compounds, which exhibit wide health benefits through their antioxidant properties, thereby preventing the development of chronic diseases [22]. As shown in Figure 2, the free TPC and bound TPC of millet sprouts showed a trend of first increasing and then decreasing with the increase in the Na_2_SeO_3_ soaking concentration. At a Na_2_SeO_3_ soaking concentration of 60 mg/L, the free and bound TPC reached the maximum value, which increased by 20.70% and 28.50%, respectively, compared with the control. These results indicated that soaking with an appropriate concentration of Na_2_SeO_3_ had a positive effect on the production of phenolic compounds in millet sprouts and displayed greater enhancement on bound TPC. It has been reported that an appropriate concentration of Na_2_SeO_3_ also increased the TPC of wheat microgreen, which may be the result of an increase in the activity of phenylalanine ammonialyase (PAL) [18]. J. Huang et al. also reported that a suitable concentration of Na_2_SeO_3_ can promote the production of polyphenols in sprouted black beans [20]. Therefore, appropriate Na_2_SeO_3_ concentrations are important for promoting phenolic accumulation in plant species. Excessive Se may also produce toxic effects on plants and affect their growth and development [3].

Flavonoids are a unique class of phenolic compounds and are important bioactive ingredients in millet sprouts [35]. The TFC of millet sprouts germinated in the presence of different Na_2_SeO_3_ soaking concentrations is shown in Figure 3. The TFC significantly increased at concentrations of less than 80 mg/L Na_2_SeO_3_, while 80 mg/L and 100 mg/L of Na_2_SeO_3_ soaking solutions resulted in a significant decrease in TFC when compared with the control. Guardado-Félix et al. reported that Na_2_SeO_3_ significantly increased the total isoflavone concentrations in chickpea sprouts, and the accumulation of isoflavones may be related to the increase in PAL activity induced by Na_2_SeO_3_ [17]. Y. Huang et al. (2022) reported that high concentrations of Na_2_SeO_3_ inhibited the synthesis of flavonoids in soybean sprouts [34]. Therefore, the Na_2_SeO_3_ concentration is a key factor in promoting the synthesis of flavonoids.

GABA is an important secondary metabolite in foxtail millet and plays a vital role in human health [36]. As shown in Figure 4, compared with the control, a 60 mg/L Na_2_SeO_3_ soaking concentration resulted in a significant increase in GABA (*p* < 0.05), while the rest of the Na_2_SeO_3_ concentrations did not produce a significant difference (*p* > 0.05). The GABA content of foxtail millet sprouts presoaked with 60 mg/L of Na_2_SeO_3_ reached 249.02 mg/kg, which increased 1.61-fold when compared with the control. Se-biofortification has been reported to increase the GABA content of black soybean sprouts, and the increased GABA level is believed to be attributed to the increased activity of glutamate decarboxylase (GAD) [20]. Our results indicated that 60 mg/L of Na_2_SeO_3_ could be selected as the suitable presoaking concentration for foxtail millet sprouts.

### 3.2. Screening of Spraying Na_2_SeO_3_ Concentration for Foxtail Millet Sprouts

#### 3.2.1. Effect of Na_2_SeO_3_ Spraying Concentration on Se Content of Millet Sprouts

The total, organic, and inorganic Se content and the Se-enrichment rate of the foxtail millet sprouts at different Na_2_SeO_3_ spraying concentrations are shown in Figure 5. The total, organic, and inorganic Se content of millet sprouts increased at first and then decreased with the rise of Na_2_SeO_3_ spraying concentrations, and the total Se content ranged from 2.80 mg/kg to 6.79 mg/kg. It has been reported that the total Se content of lupin sprouts was in the range of 1.13–4.92 mg/kg by spraying 2–8 mg/L of Na_2_SeO_3_ solution during germination, but when the Na_2_SeO_3_ concentration surpassed 6 mg/L, the total Se content of lupin sprouts no longer significantly increased [37]. Rao et al. also reported that the total Se content of soybean shoots was reduced by a high concentration of Na_2_SeO_3_ solution [19]. It has been reported that with selenite application, exogenous inorganic Se is taken up by roots through passive diffusion, then accumulated and transformed into organic Se [15]. The application of high concentrations of Na_2_SeO_3_ during seed germination increases the density of endodermal cells in roots, thereby inhibiting the growth of both hypocotyls and roots [38]. Therefore, the decrease in Se content may be due to the damage caused by high concentration of Na_2_SeO_3_ to roots, thus affecting the enrichment of Se [39].

#### 3.2.2. Effects of Na_2_SeO_3_ Spraying Concentration on Phenolic and GABA Content

Figure 6 shows the effects of different Na_2_SeO_3_ spraying concentrations on the free and bound TPC of millet sprouts. Similar to the results of the Na_2_SeO_3_ presoaking treatment, free and bound TPC showed first an increasing and then a decreasing trend with the increase in the Na_2_SeO_3_ spraying concentration. Compared to the control, both the free and bound TPC reached maximum values under the 2 mg/L Na_2_SeO_3_ spraying treatment and were increased by 25.24% and 23.45%, respectively. As shown in Figure 7 and Figure 8, the TFC and GABA content significantly increased by 28.09% and 53.31%, respectively, only at the Na_2_SeO_3_ spraying concentration of 2 mg/L. It was indicated that a low concentration of Na_2_SeO_3_ spraying solution can promote the accumulation of polyphenols and GABA, which may also be attributed to the increased activity of PAL and GAD. In this study, 2 mg/L of Na_2_SeO_3_ solution was selected as the suitable spraying concentration for the germination process.

Our results showed that the Na_2_SeO_3_ soaking treatment was far superior to the Na_2_SeO_3_ spraying treatment in promoting Se levels and the accumulation of organic Se. However, the Na_2_SeO_3_ spraying treatment was equally effective as the Na_2_SeO_3_ soaking treatment in promoting phenolic and GABA synthesis in foxtail millet sprouts.

### 3.3. Change in Se Content During Se-Enriched Germination of Foxtail Millet

As shown in Table 1, the application of exogenous Na_2_SeO_3_ significantly increased the total and organic Se content of foxtail millet sprouts. In terms of organic Se, the Na_2_SeO_3_ soaking treatment exhibited a significantly higher enrichment effect than the Na_2_SeO_3_ spraying treatment at all the germination stages. Similarly, the soaking treatment caused a significantly higher Se-enrichment rate than the spraying treatment. The Se-enrichment rate exhibited a gradual rising tendency during the germination process, indicating that the germination process could effectively convert inorganic Se into organic Se. Seeds can maintain high vitality during the germination stage and remain sensitive to Se uptake and transformation. However, Se is not necessary for plants, and excessive exogenous inorganic Se could cause damage to plants. Therefore, based on self-protection, germinated seeds should rapidly convert inorganic Se to organic Se through various metabolic pathways [27].

The organic Se content of millet sprouts with the Na_2_SeO_3_ presoaking treatment decreased at the fourth germination day (G4), which may be due to the partial organic Se conversion to dimethyl-selenide (DMSe) or dimethyl-diselenide (DMDSe) and volatilization caused by the sulfur metabolism pathway, so as to reduce the toxic effect of excessive Se [40]. Puccinelli et al. also showed that the total Se content of the leaves, stems, and roots of basil plants decreased with a prolonged culture time [41]. Therefore, an appropriate Se treatment and germination period are key factors for the production of high-quality Se-enriched foxtail millet sprouts. Furthermore, the proper growth stage of the sprouts for harvest is equally important, as this is directly correlated to the readiness of the edible portion of sprouts [5]. According to the Se content of the edible parts of foxtail millet sprouts, it can be concluded that the Na_2_SeO_3_ soaking treatment was more suitable for the production of Se-enriched millet sprouts.

### 3.4. Changes in TPC During Se-Enriched Germination of Foxtail Millet

The changes in the free and bound TPC of millet sprouts during the Se-enriched germination process are shown in Figure 9. The bound TPC of the millet sprouts decreased slightly during the first germination day (G1) and the second day (G2) but increased rapidly after three germination days. The free TPC slightly reduced at the G1 stage without a statistical difference, compared with ungerminated seeds (G0) (*p* > 0.05), and then increased significantly (*p* < 0.05) with the extension of the germination process, which is in agreement with our previous study [24]. Germination contributed to the accumulation of phenolic compounds, which was ascribed to the PAL activated during the germination process [29]. The phenolic acids bound to non-starch polysaccharides in the grain cell wall could be released under the action of the cell wall degrading enzyme activated during germination, which would cause a rapid increase in free TPC [42].

During early germination stages (G1 and G2), both the bound and free TPC of the millet sprouts displayed no significant difference (*p* > 0.05) between the Na_2_SeO_3_ soaking or spraying treatments and the control. In comparison, during the later stages of germination (G3 and G4), the bound and free TPC of the two Na_2_SeO_3_ treatments were significantly higher (*p* < 0.05) than those of the control. For the bound TPC, the highest level of 462.87 mg FAE/100 g was observed in foxtail millet sprouts with the Na_2_SeO_3_ spraying treatment at the G4 stage, followed by the Na_2_SeO_3_ soaking treatment, and both of them were significantly higher (*p* < 0.001) than the control. As for the free phenolics, compared with the control, the Na_2_SeO_3_ soaking treatment showed a significant increase (*p* < 0.05) at G4, reaching 437.65 mg FAE/100 g, which was superior to the Na_2_SeO_3_ spraying treatment.

A similar phenomenon was observed in soybean sprouts; after being presoaked with Na_2_SeO_3_, the TPC of soybean sprouts exhibited a slight increase during the initial germination stages, but increased dramatically after three days of germination, compared to the control group [34]. It was indicated that the effect of Na_2_SeO_3_ on the TPC of millet sprouts is not only related to the method of Na_2_SeO_3_ treatment but also to the germination stage.

### 3.5. Changes in TFC During Se-Enriched Germination of Foxtail Millet

The changes in the TFC of millet sprouts during Se-enriched germination are shown in Figure 10. The TFC of foxtail millet sprouts with all the different treatments showed first a decreasing and then an increasing trend during the germination process. The early decrease in the TFC was probably due to the leaching of water-soluble flavonoid glycosides and the activation of polyphenol oxidase and peroxidase, while the increased TFC after two germination days was attributed to the activation of flavonoid synthetase [29].

Consistent with the performance of the free and bound TPC, the TFC of millet sprouts did not show a significant difference between the Na_2_SeO_3_ treatment groups and the control during the early germination stages. After three days of germination, the TFC of the millet sprouts with Na_2_SeO_3_ soaking and spraying treatments were significantly higher (*p* < 0.001) than the control. However, after four days of germination, only the Na_2_SeO_3_ soaking treatment group showed a significantly higher (*p* < 0.05) TFC than the control, reaching the highest value of 76.56 mg RE/100 g of DW.

Guardado-Félix et al. observed that the total isoflavonoid concentration of Se-enriched chickpea sprouts was far higher than that of the control group during the later stages of germination, which indicates that the enrichment effect of Na_2_SeO_3_ treatment on flavonoids was also related to the seed germination cycle [17]. Similar effects regarding the enhancement of flavonoid biosynthesis in germinated seeds by Na_2_SeO_3_, such as radish sprouts [16], black soybean sprouts [20], and buckwheat sprouts [43], have been detected. It has been reported that Na_2_SeO_3_ enhances flavonoid accumulation by increasing the expression levels of flavonoid biosynthesis structural genes [10].

### 3.6. Changes in GABA Content During Se-Enriched Germination of Foxtail Millet

The changes in the GABA content of the foxtail millet with three different treatments during germination are shown in Figure 11. The GABA levels of millet sprouts with both Na_2_SeO_3_ treatments were significantly higher (*p* < 0.001) than that of the control at the same germination stage, separately. The Na_2_SeO_3_ spraying treatment displayed a superior accumulative effect on GABA content compared to the Na_2_SeO_3_ soaking treatment at the G2 and G3 stages. However, after four days of germination, the GABA of the Na_2_SeO_3_ soaking sprouts showed the highest level of 254.35 mg/kg of DW, followed by the Na_2_SeO_3_ spraying treatment.

It has been reported that excessive concentrations of selenite increases reactive oxygen species (ROS) levels in peanut sprouts [38]; thus, a high concentration of exogenous Na_2_SeO_3_ causes a similar effect as an abiotic stress [15], while increased GABA in germinated seeds helps to enhance the antioxidant defense system and reduce ROS levels under stress conditions, thereby protecting plants from oxidative damage [44]. Given the limited research on the impact of Na_2_SeO_3_ on GABA of germinated seeds, our study clearly indicates that the appropriate concentration of Na_2_SeO_3_ soaking or spraying can effectively induce the accumulation of GABA in germinating seeds.

### 3.7. Changes in Individual Phenolic Compounds of Foxtail Millet Sprouts During Se-Enriched Germination

Typical phenolic profiles of the free and bound extracts from the Se-enriched foxtail millet sprouts are provided in Appendix A. The phenolic compounds identified in Se-enriched foxtail millet sprouts were consistent with our previous research [24], and their characteristic data of retention time (RT), MS ([M-H]^−^), and MS/MS data, and the identification results of free and bound phenolic compounds are listed in Appendix A and Appendix A, respectively. Foxtail millet sprouts with Na_2_SeO_3_ soaking or spraying treatment exhibited similar phenolic profiles with the control at the same germination period, with a typical chromatogram shown in Figure 12A,B.

#### 3.7.1. Free Phenolic Compounds

The content changes in the individual free phenolic compounds during Se-enriched germination are presented in Appendix A. The predominant phenolic compounds of free phenolic fractions were feruloylquinic acid, 4-p-coumarinoylquinic acid, and 3,7-dimethylquercetin, which were rapidly accumulated during the germination process. N-feruloyl serotonin and N-(p-coumaroyl) serotonin emerged during the second day and accumulated gradually with the prolongation of the germination process. These results were consistent with our previous study [24]. In this study, feruloylquinic acid and 4-p-coumaroylquinic acid reached their highest levels at the G3 stage, while 3,7-dimethylquercetin reached its highest level at the G4 stage. These three major free phenolic compounds were significantly increased (*p* < 0.05) by the two Se-enriched treatments. Feruloylquinic acid and 4-p-coumaroylquinic acid displayed the highest levels of 18.59 mg/kg of DW and 14.02 mg/kg of DW, respectively, through the Na_2_SeO_3_ spraying treatment. In comparison, the Na_2_SeO_3_ soaking treatment resulted in the highest level for 3,7-dimethylquercetin, reaching 25.70 mg/kg of DW.

In addition, both Na_2_SeO_3_ treatments also contributed to the accumulation of N-feruloyl serotonin and N-(p-coumaroyl) serotonin. The content of p-hydroxybenzoic acid was increased significantly through the Na_2_SeO_3_ soaking treatment, with the highest increase observed during the later stages of germination, and the highest concentration reached was approximately 2.86-fold higher than that of the control. The content of 1-O-p-coumaroyl-3-O-feruloylglycerol was significantly increased by Na_2_SeO_3_ soaking and spraying during the later stages of germination. Hydroxycinnamic acid amides (HCAAs) are a unique class of phenolic compounds found in foxtail millet sprouts, including N′-caffeoylspermidine, N′-p-coumaroyl-N″-feruloylspermidine, N′, N″-di-p-coumaroylspermidine, and N′, N″-di feruloylspermine. Most of the HACC levels of Se-enriched foxtail millet sprouts were not elevated or decreased compared to the control, with apigenin-C-pentosyl-C-hexoside, trans-p-coumaric acid, and trans-ferulic acid exhibiting similar performance.

#### 3.7.2. Bound Phenolic Compounds

The content changes in the individual bound phenolic compounds during Se-enriched germination are presented in Appendix A. The concentrations of the bound phenolic compounds were mostly enhanced by Na_2_SeO_3_ soaking and spraying treatments, particularly during the later stages of germination. Among them, trans-p-coumaric acid and trans-ferulic acid were the most abundant bound phenolics in foxtail millet sprouts, and their levels increased with the prolongation of the germination process. Trans-p-coumaric acid and trans-ferulic acid in foxtail millet sprouts reached the highest levels at the G4 stage and exhibited the highest increases through the Na_2_SeO_3_ spraying treatment, which increased by 97.40% and 26.24%, respectively. Similar results of Na_2_SeO_3_ treatment increasing levels of ferulic acid and *p*-coumaric acid have been observed in rice sprouts [15] and black bean sprouts [20]. In addition, during the later stages of germination, protocatechuic acid, *p*-hydroxybenzaldehyde, vanillic acid, syringic acid, 3,7-dimethylquercetin, ferulic acid dimers (DFAs), and ferulic acid trimers (TFAs) also increased to varying degrees by Na_2_SeO_3_ soaking or spraying treatments. However, *cis*-ferulic acid significantly (*p* < 0.05) reduced with the Na_2_SeO_3_ spraying treatment during germination, which showed the opposite result compared with the Na_2_SeO_3_ soaking treatment.

Overall, both Na_2_SeO_3_ treatments promoted the enrichment of most phenolic acids in millet sprouts during the later stages of germination. Phenolic acids play an important role in plant resistance to biological and abiotic stresses due to their powerful antioxidant capability [45]. The higher enrichment of these phenolic acids may be attributed to defending the stress of a high-concentration Na_2_SeO_3_ solution. Although the two Na_2_SeO_3_ treatments differed in the regulation of some phenolic components in foxtail millet sprouts, the effects on the enrichment of the predominate free and bound phenolic compounds were positive. Phenolic acids are mainly produced via the phenylpropanoid pathway, which could be mediated by proper Na_2_SeO_3_ treatment through upregulating PAL, trans-cinnamate-4-hydroxylase, and cinnamyl-alcohol dehydrogenase [38].

### 3.8. Changes in Antioxidant Activity of Foxtail Millet Sprouts During Se-Enriched Germination

The antioxidant activity of phenolic extracts was evaluated by DPPH, ABTS, and FRAP assays. As shown in Table 2, germination with or without Na_2_SeO_3_ treatment caused the enhancement of the antioxidant capacity of foxtail millet sprouts. Compared with the control, Se-enriched millet sprouts exhibited significantly higher (*p* < 0.05) antioxidative capacity.

Regarding the DPPH radical scavenging activity, DPPH values of the phenolics from Se-enriched foxtail millet sprouts showed no significant difference (*p* > 0.05) from the control at the early germination stages. However, during the later stages (G3 and G4), these values were significantly higher (*p* < 0.05) than those of the control. Among them, the phenolics of millet sprouts with Na_2_SeO_3_ spraying treatment displayed the highest DPPH value at the G4 stage, reaching 15.30 μmol TE/g.

Unlike the DPPH assay, the ABTS values of phenolics from Se-enriched millet sprouts were significantly higher (*p* < 0.05) than those of the control during the initial three days of germination, but there was no significant difference (*p* > 0.05) at the G4 stage when comparing the control. The results of various antioxidant assays showed some variability, which may be attributed to the distinct mechanisms of action inherent to each assay.

For ferric-reducing antioxidant power (FRAP), similar to the DPPH assay, the FRAP values of phenolics from the foxtail millet sprouts with the two Na_2_SeO_3_ treatments were observed to be significantly higher (*p* < 0.05) than those of the control during the later germination stages. The millet sprouts with Na_2_SeO_3_ spraying treatment exhibited the highest FRAP value at the G4 stage, reaching 21.76 μmol TE/g.

Overall, the Na_2_SeO_3_ soaking or spraying treatments enhanced the antioxidant activity of foxtail millet sprouts, which was mainly attributed to the accumulation and synthesis of diverse polyphenols [17]. It has been reported that phenolic extracts from Se-enriched black soybean sprouts have higher antioxidant activities than those of germinated black soybean sprouts without Se treatment [20].

## 4. Conclusions

This study demonstrated that soaking and spraying with Na_2_SeO_3_ significantly influenced the Se content of foxtail millet sprouts, with the Na_2_SeO_3_ soaking treatment exhibiting superior Se-enrichment effectiveness. The appropriate concentration of Na_2_SeO_3_ can increase the TPC, TFC, and GABA content of foxtail millet sprouts. The main phenolic compounds (such as trans-ferulic acid, trans-p-coumaric acid, and feruloylquinic acid) and serotonin derivatives were further enriched by two Na_2_SeO_3_ treatments. The antioxidant activity of Se-enriched millet sprouts was further enhanced compared to the general foxtail millet sprouts. Foxtail millet is one of the world’s most important grains. The application of Se-biofortification can increase the levels of polyphenols, GABA, and antioxidant activity during the germination process, which provides certain ideas for the development of high-quality and high-added value Se-enriched functional food. Nevertheless, the form in which organic Se exists following Se-enriched treatments of foxtail millet sprouts, as well as its bioavailability, remains unclear. Further research is required to develop foxtail millet sprouts as a functional Se-enriched food.

## Figures and Tables

**Figure 1 foods-13-03916-f001:**
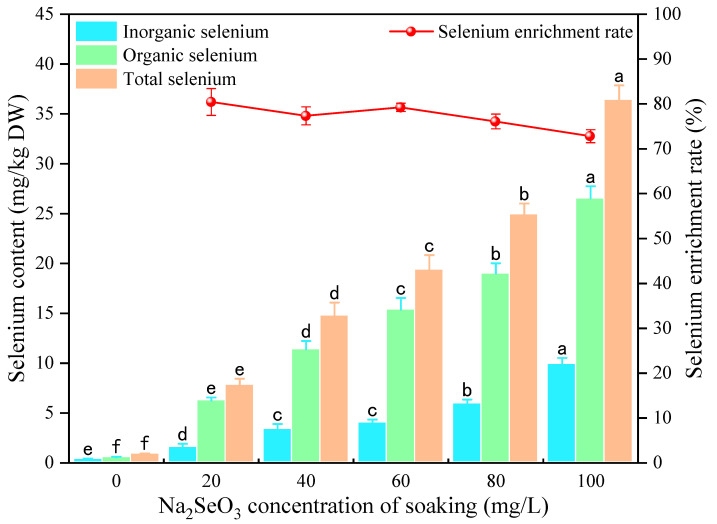
The effects of different concentrations of Na_2_SeO_3_ soaking solution on the total, organic, and inorganic Se content and the Se-enrichment rate of millet sprouts. Note: The different letters represent the difference in significance under different Na_2_SeO_3_ soaking concentrations, *p*-value < 0.05.

**Figure 2 foods-13-03916-f002:**
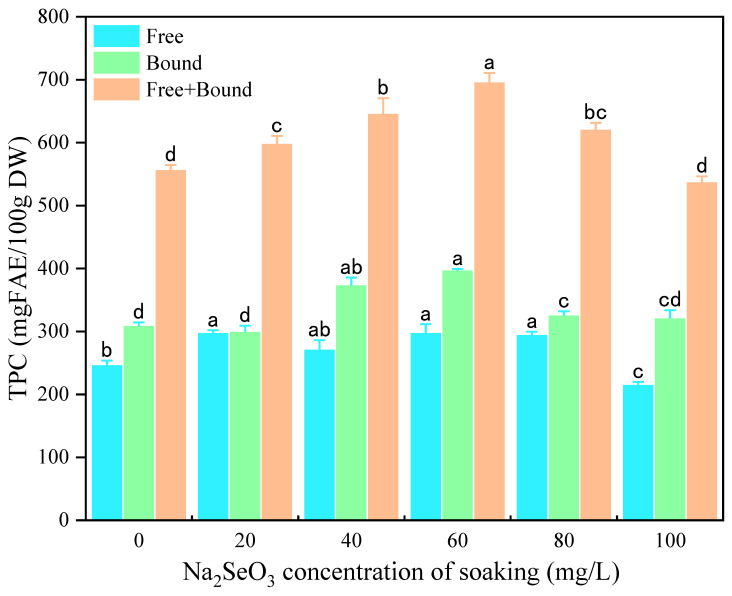
The effect of different concentrations of Na_2_SeO_3_ soaking solution on the TPC of millet sprouts. Note: The different letters represent the difference in significance under different Na_2_SeO_3_ soaking concentrations, *p*-value < 0.05.

**Figure 3 foods-13-03916-f003:**
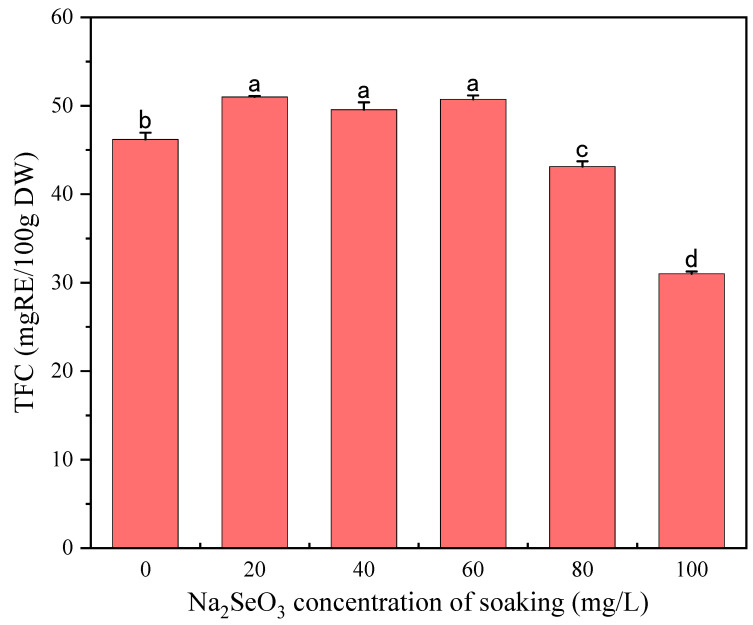
The effect of different concentrations of Na_2_SeO_3_ soaking solution on the TFC of millet sprouts. Note: The different letters represent the difference in significance under different Na_2_SeO_3_ soaking concentrations, *p*-value < 0.05.

**Figure 4 foods-13-03916-f004:**
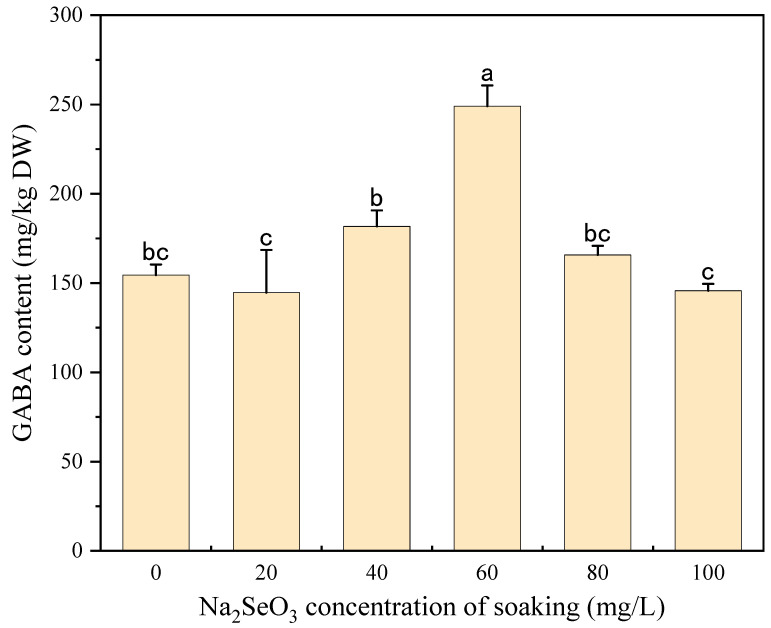
The effect of different concentrations of Na_2_SeO_3_ soaking solution on the GABA content of millet sprouts. Note: The different letters represent the difference in significance under different Na_2_SeO_3_ soaking concentrations, *p*-value < 0.05.

**Figure 5 foods-13-03916-f005:**
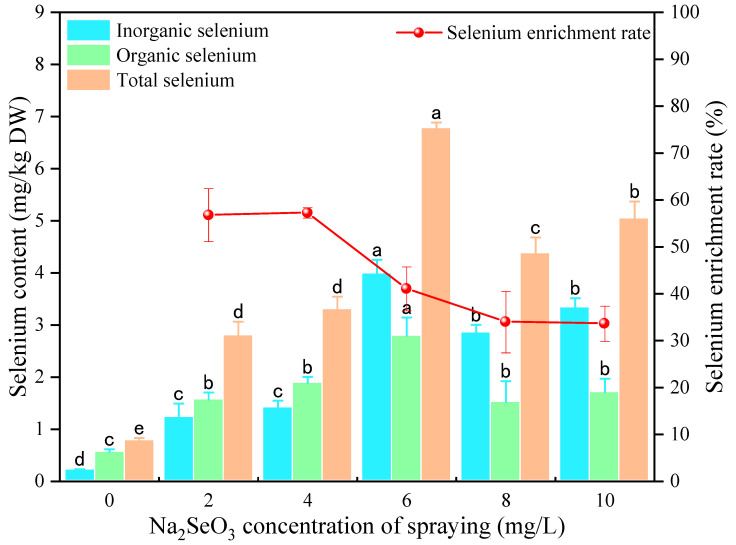
The effects of different concentrations of Na_2_SeO_3_ spraying solution on the total, organic, and inorganic Se content and the Se-enrichment rate of millet sprouts. Note: The different letters represent the difference in significance under different Na_2_SeO_3_ spraying concentrations, *p*-value < 0.05.

**Figure 6 foods-13-03916-f006:**
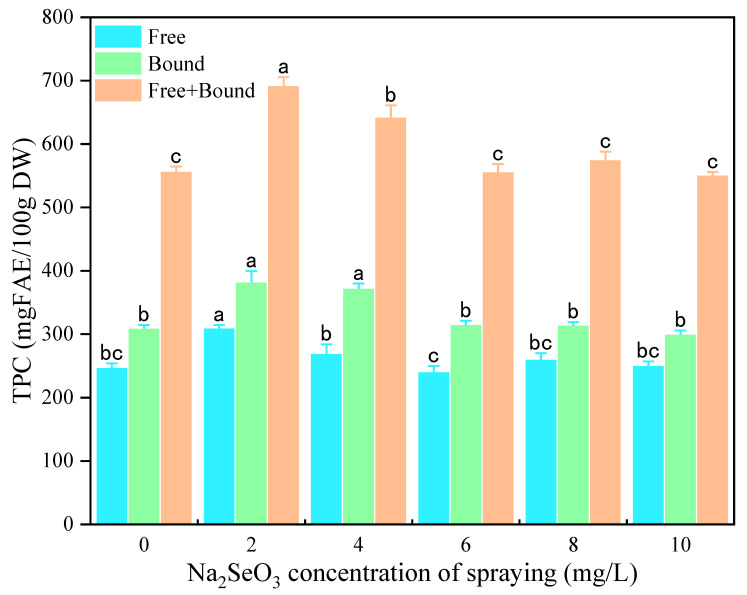
The effect of different concentrations of Na_2_SeO_3_ spraying solution on the TPC of millet sprouts. Note: The different letters represent the difference in significance under different Na_2_SeO_3_ soaking concentrations, *p*-value < 0.05.

**Figure 7 foods-13-03916-f007:**
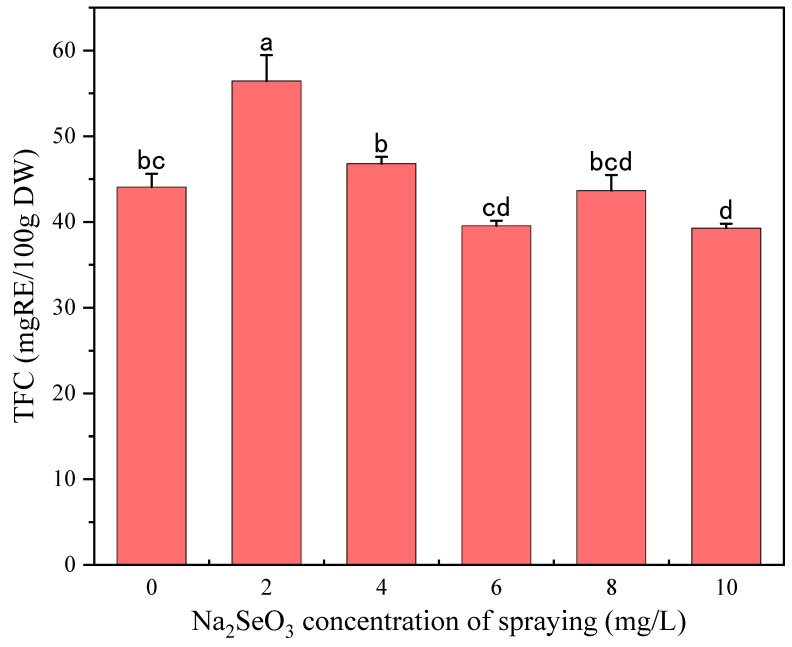
The effect of different concentrations of the Na_2_SeO_3_ spraying solution on the TFC of millet sprouts. Note: The different letters represent the difference in significance under different Na_2_SeO_3_ soaking concentrations, *p*-value < 0.05.

**Figure 8 foods-13-03916-f008:**
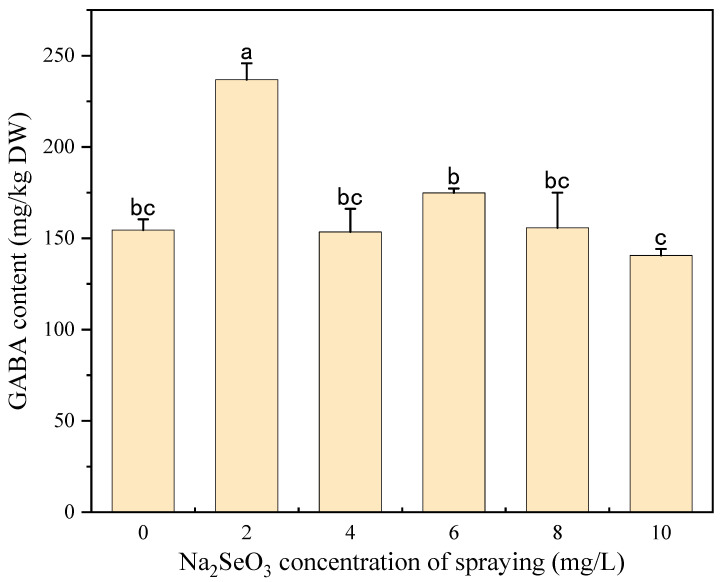
The effect of different concentrations of the Na_2_SeO_3_ spraying solution on the GABA content of millet sprouts. Note: The different letters represent the difference in significance under different Na_2_SeO_3_ spraying concentrations, *p*-value < 0.05.

**Figure 9 foods-13-03916-f009:**
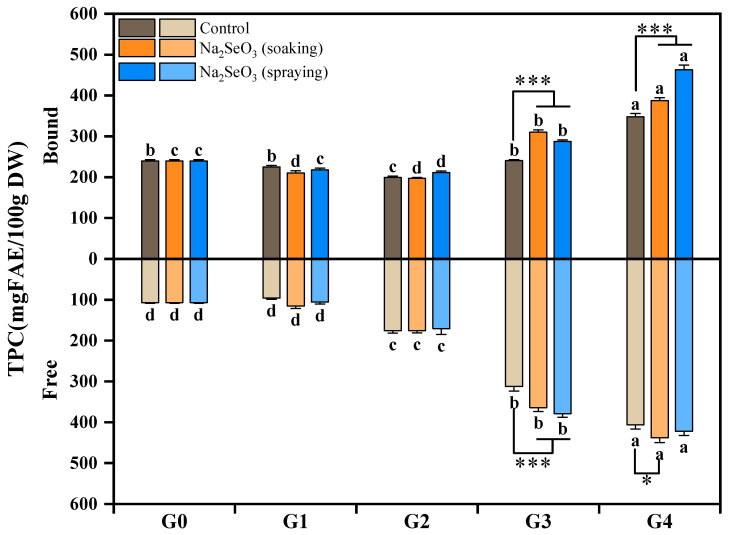
Dynamic changes in TPC of foxtail millet during Se-enriched germination. Different lower cases indicate significant differences at different germination days (*p* < 0.05), and asterisks indicate statistically significant differences between Na_2_SeO_3_ treatment and control (* *p* < 0.05, *** *p* < 0.001).

**Figure 10 foods-13-03916-f010:**
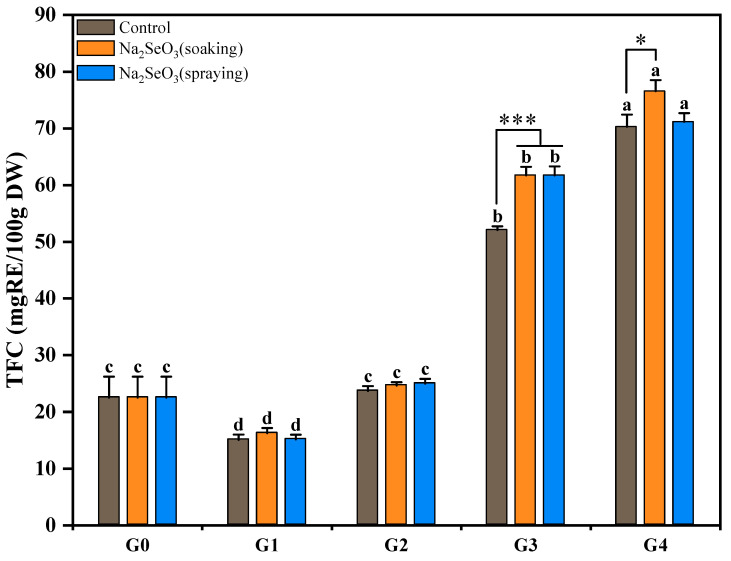
Dynamic changes in TFC of foxtail millet during Se-enriched germination. Different lower cases indicate significant differences at different germination days (*p* < 0.05), and asterisks indicate statistically significant differences between Na_2_SeO_3_ treatment and control (* *p* < 0.05, *** *p* < 0.001).

**Figure 11 foods-13-03916-f011:**
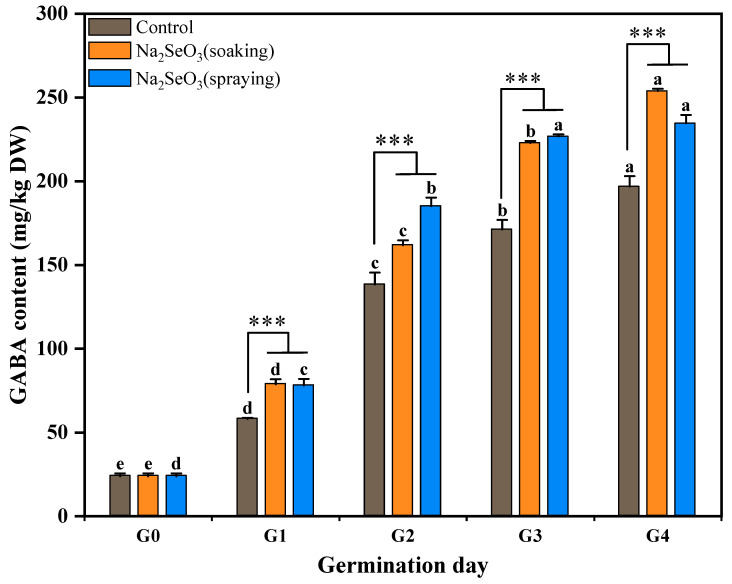
Dynamic changes in GABA content of foxtail millet during Se-enriched germination. Different lower cases indicate significant differences at different germination days (*p* < 0.05), and asterisks indicate statistically significant differences between Na_2_SeO_3_ treatment and control (*** *p* < 0.001).

**Figure 12 foods-13-03916-f012:**
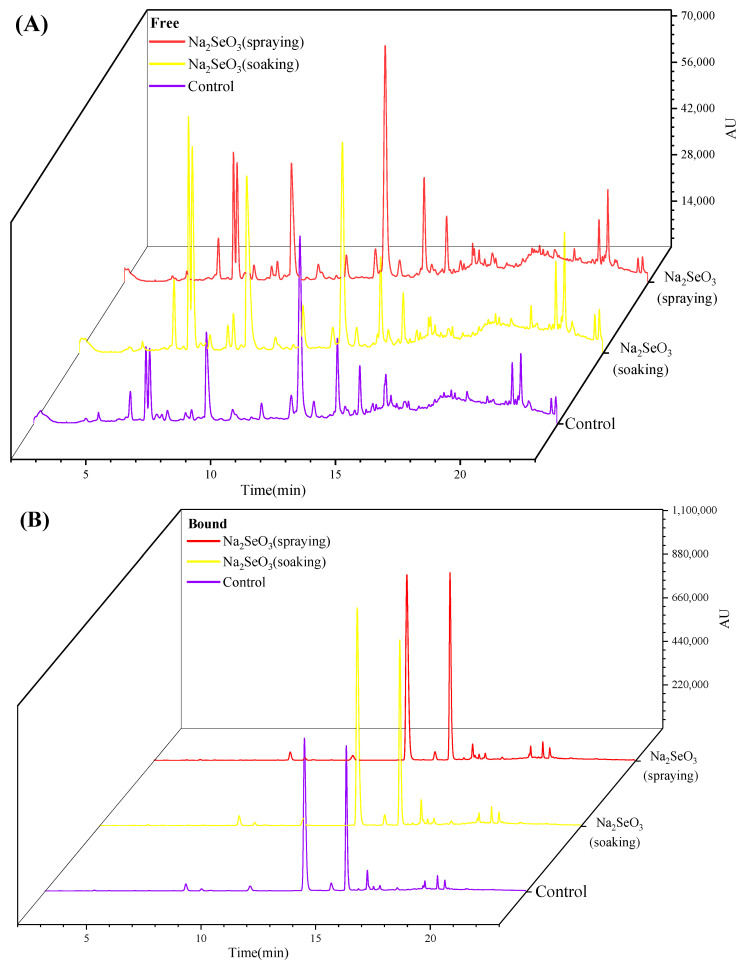
UPLC chromatograms of free (**A**) and bound (**B**) phenolics from foxtail millet sprouts after germination for three days. The detection wavelengths of free and bound phenolics were 320 nm and 280 nm, respectively.

**Table 1 foods-13-03916-t001:** Effects of Na_2_SeO_3_ treatments on Se content (mg/kg DW) of foxtail millet sprouts during germination process.

Se Content (mg/kg DW)/ Se-Enrichment Rate (%)	Germination Treatment	Germination Stage			
		G1	G2	G3	G4
Total Se	Control	0.81 ± 0.02 ^Ac^	0.87 ± 0.01 ^Ac^	0.83 ± 0.05 ^Ac^	1.02 ± 0.06 ^Ac^
	Na_2_SeO_3_ (soaking)	12.58 ± 0.18 ^Aa^	12.60 ± 0.25 ^Aa^	12.89 ± 0.44 ^Aa^	10.32 ± 0.39 ^Ba^
	Na_2_SeO_3_ (spraying)	4.16 ± 0.03 ^Bb^	4.33 ± 0.02 ^Bb^	5.35 ± 0.04 ^Ab^	5.01 ± 0.06 ^Ab^
Organic Se	Control	0.56 ± 0.03 ^Ac^	0.61 ± 0.14 ^Ac^	0.54 ± 0.06 ^Ac^	0.64 ± 0.08 ^Ad^
	Na_2_SeO_3_ (soaking)	8.81 ± 0.26 ^Ca^	10.11 ± 0.20 ^Ba^	10.97 ± 0.44 ^Aa^	8.37 ± 0.45 ^Cc^
	Na_2_SeO_3_ (spraying)	2.21 ± 0.15 ^Bb^	2.71 ± 0.09 ^Bb^	3.23 ± 0.18 ^Ab^	3.43 ± 0.11 ^Ab^
Inorganic Se	Control	0.25 ± 0.05 ^Ac^	0.26 ± 0.12 ^Ac^	0.28 ± 0.06 ^Ac^	0.37 ± 0.04 ^Ac^
	Na_2_SeO_3_ (soaking)	3.77 ± 0.10 ^Aa^	2.48 ± 0.07 ^Ba^	1.92 ± 0.08 ^Cb^	1.95 ± 0.06 ^Ca^
	Na_2_SeO_3_ (spraying)	1.94 ± 0.15 ^Bb^	1.62 ± 0.07 ^Bb^	2.12 ± 0.14 ^Aa^	1.58 ± 0.08 ^Bb^
Se-enrichment rate	Control	-	-	-	-
	Na_2_SeO_3_ (soaking)	70.02	80.31	85.10	81.01
	Na_2_SeO_3_ (spraying)	53.21	62.53	60.35	68.44

Different capital letters indicate significant differences in different germination days (*p* < 0.05), while different lowercase letters indicate significant differences in different germination treatments (*p* < 0.05).

**Table 2 foods-13-03916-t002:** Effects of Na_2_SeO_3_ treatments on antioxidant activity (μmol TE/g) of foxtail millet sprouts during germination process.

Antioxidant Assay	Germination Treatment	Germination Stage			
		G1	G2	G3	G4
DPPH	Control	5.97 ± 0.60 ^Ca^	5.28 ± 0.49 ^Ca^	8.95 ± 0.55 ^Bb^	12.86 ± 0.40 ^Ab^
	Na_2_SeO_3_ (soaking)	5.50 ± 0.29 ^Da^	6.64 ± 0.08 ^Ca^	11.76 ± 0.23 ^Ba^	14.90 ± 0.23 ^Aa^
	Na_2_SeO_3_ (spraying)	6.16 ± 0.23 ^Ca^	6.07 ± 0.42 ^Ca^	11.64 ± 0.50 ^Ba^	15.30 ± 0.40 ^Aa^
ABTS	Control	6.45 ± 0.25 ^Db^	8.92 ± 0.17 ^Cb^	12.22 ± 0.26 ^Bb^	13.78 ± 0.39 ^Aa^
	Na_2_SeO_3_ (soaking)	8.03 ± 0.29 ^Da^	9.69 ± 0.24 ^Cab^	13.42 ± 0.18 ^Ba^	14.25 ± 0.36 ^Aa^
	Na_2_SeO_3_ (spraying)	7.62 ± 0.18 ^Da^	9.72 ± 0.29 ^Ca^	12.85 ± 0.45 ^Bab^	14.42 ± 0.32 ^Aa^
FRAP	Control	7.86 ± 0.22 ^Da^	8.40 ± 0.09 ^Ca^	13.95 ± 0.10 ^Bc^	19.31 ± 0.16 ^Ab^
	Na_2_SeO_3_ (soaking)	7.85 ± 0.12 ^Da^	8.76 ± 0.06 ^Ca^	17.03 ± 0.06 ^Ba^	21.49 ± 0.29 ^Aa^
	Na_2_SeO_3_ (spraying)	7.69 ± 0.14 ^Ca^	7.77 ± 0.19 ^Cb^	15.96 ± 0.11 ^Bb^	21.76 ± 0.14 ^Aa^

Different capital letters indicate significant differences in different germination days (*p* < 0.05), while different lowercase letters indicate significant differences in different germination treatments (*p* < 0.05). Results are expressed as mean ± SD.

## Data Availability

The original contributions presented in the study are included in the article; further inquiries can be directed to the corresponding author.

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
