# Peer review of "Effects of Sodium Selenite on Accumulations of Selenium and GABA, Phenolic Profiles, and Antioxidant Activity of Foxtail Millet During Germination"

_foods, 2024, doi:10.3390/foods13233916_

Round 1

Reviewer 1 Report

Comments and Suggestions for Authors

General comments:  Manuscript is a wide overview of how applications of selenium ions impact a grain crop of some importance mainly in Asia.  The focus is on the effects on the seed although it is used as forage grass in some areas.  Two methods of applications were compared.  Is there a concern for accumulation of excess Se using these types of treatments?  The study seems adequately described.

Line 36:  Change to read “Edible plants are believed to be…”

Line 37:  Change to “lots of” to “many”

Line 39:  Change “for” to “due to”

Line 44:  Add “The” before “germination”

Lines 51 to 52:  Change to read “Anthocyanins were effectively…”

Lines 51 to 53:  Is the correct reference cited here?  Reference 9 is discussing mung bean, not radish.

Line 89:  Change to read “…samples were provided…”

Line 146:  Reword to read “…derivatives without available standards…”

Reference format:  Check instructions to authors.  Article titles should only have first word and proper nouns capitalized.  Check 3, 6, 7, 8, 18, 22, 23, 25, 26, 27, 33 and 35.

Reviewer 2 Report

Comments and Suggestions for Authors

Dear authors,

congratulations for the good concept of the study, clear and concise results and reasonable discussion. There is no need for revision.

Sincerely,

Author Response

comment: Dear authors,

congratulations for the good concept of the study, clear and concise results and reasonable discussion. There is no need for revision.

response: Thanks for your positive evaluation on this paper.

Reviewer 3 Report

Comments and Suggestions for Authors

Please provide the brand and origin details of each instrument or equipment in the materials and methods section.

In section 2.5 specify the UPLC conditions for the GABA content. Although you mentioned that you used a previously described method by your group, readers should have easy and quick access to these details of your current manuscript in the file. Same for section 2.6., and 2.7

Sentence in line 169 the word increase is repetitive.

Selenium in Figure 3 should be capitalized.

For easier understanding of Table 1 I suggest you put a line under each germination treatment

Results are described properly.

In Figure 8, it might enrich your paper if you identify at least the most predominant peaks

Supplementary Table 3 is very close to Table S2

Results in Table S3 and S4 are, in my opinion, also very important, please analyse if it's pertinent to include them in the main manuscript

Reviewer 4 Report

Comments and Suggestions for Authors

1- Material and method section: the following paragraph is 100% to previously published paper: Xiang et al., 2023

Modification on phenolic profiles and enhancement of antioxidant activity of proso millets during germination

The polyphenol standards, including caffeic acid, chlorogenic acid, protocatechualdehyde, p -hydroxybenzaldehyde, p -hydroxybenzoic acid, p -coumaric acid, protocatechuic acid, ferulic acid, vanillic acid and rutin, were acquired from Shanghai Yuanye Biotechnology Co., Ltd. (Shanghai, China). All the other regular chemicals, including sodium carbonate (Na2 CO3 ), ferric trichloride (FeCl3 ), sodium hydroxide (NaOH), aluminum trichloride (AlCl3 ), sodium nitrite (NaNO2 ) ethyl acetate and N -hexane, were acquired from Tianjin Deen Chemical Reagent Co., Ltd. (Tianjin,

2- The authors are invited to verify the list of abbreviation; FAF, DW, TFC, TPC..... reported only in Figures 

3- Figure 2: why A, B and C it will be better if you separate as Figure 2, Figure 3..... and rectify the citation of figures in the main text 

4-Table 1 and Table 2: letters of test significance should be written as superscript (exponent) . Actually, not uniform 

5- Figure 5, 6 and 7: letters of test significance are written as Capital why?

Reviewer 5 Report

Comments and Suggestions for Authors

The article is interesting; however, the experimental design requires further details. Authors state all experiments were carried out by triplicate; however this is insufficient to reflect the complexity of plants. The UPLC-MS/MS profile indicates the identification of components; however, only some compounds were confirmed by comparison with commercial standards. Authors must specify the identification level, since they only provide the maximum wavelength and the m/z of the molecular ion, without providing the fragments used for identification. Moreover, several components identified without standards are proposed with specific positions that cannot be elucidated. 

Round 2

Reviewer 3 Report

Comments and Suggestions for Authors

Dear authors,

The are no more comments on my behalf.

Kind regards

Author Response

Thanks for the reviewer's patience.